# Mortality Predictors in Severe SARS-CoV-2 Infection

**DOI:** 10.3390/medicina58070945

**Published:** 2022-07-18

**Authors:** Mihai Lazar, Ecaterina Constanta Barbu, Cristina Emilia Chitu, Ana-Maria-Jennifer Anghel, Cristian-Mihail Niculae, Eliza-Daniela Manea, Anca-Cristina Damalan, Adela-Abigaela Bel, Raluca-Elena Patrascu, Adriana Hristea, Daniela Adriana Ion

**Affiliations:** 1Faculty of Medicine, University of Medicine and Pharmacy Carol Davila, No. 37, Dionisie Lupu Street, Sector 2, 020021 Bucharest, Romania; ecaterina.barbu@umfcd.ro (E.C.B.); cristinaemilia_tisu@yahoo.com (C.E.C.); cristian.niculae@drd.umfcd.ro (C.-M.N.); eliza.manea@drd.umfcd.ro (E.-D.M.); raluca.jipa1@drd.umfcd.ro (R.-E.P.); adriana.hristea@umfcd.ro (A.H.); danielaion7@ymail.com (D.A.I.); 2National Institute for Infectious Diseases Prof. Dr. Matei Bals, No. 1, Calistrat Grozovici Street, District 2, 021105 Bucharest, Romania; amya.anghel@gmail.com (A.-M.-J.A.); scoicaru.anca@gmail.com (A.-C.D.); bel.adela@yahoo.com (A.-A.B.)

**Keywords:** SARS-CoV-2, COVID-19, risk factor, mortality, prediction score, quantitative evaluation, density clusters

## Abstract

*Background and Objectives*: The severe forms of SARS-CoV-2 pneumonia are associated with acute hypoxic respiratory failure and high mortality rates, raising significant challenges for the medical community. The objective of this paper is to present the importance of early quantitative evaluation of radiological changes in SARS-CoV-2 pneumonia, including an alternative way to evaluate lung involvement using normal density clusters. Based on these elements we have developed a more accurate new predictive score which includes quantitative radiological parameters. The current evolution models used in the evaluation of severe cases of COVID-19 only include qualitative or semi-quantitative evaluations of pulmonary lesions which lead to a less accurate prognosis and assessment of pulmonary involvement. *Materials and Methods:* We performed a retrospective observational cohort study that included 100 adult patients admitted with confirmed severe COVID-19. The patients were divided into two groups: group A (76 survivors) and group B (24 non-survivors). All patients were evaluated by CT scan upon admission in to the hospital. *Results*: We found a low percentage of normal lung densities, PaO_2_/FiO_2_ ratio, lymphocytes, platelets, hemoglobin and serum albumin associated with higher mortality; a high percentage of interstitial lesions, oxygen flow, FiO_2_, Neutrophils/lymphocytes ratio, lactate dehydrogenase, creatine kinase MB, myoglobin, and serum creatinine were also associated with higher mortality. The most accurate regression model included the predictors of age, lymphocytes, PaO_2_/FiO_2_ ratio, percent of lung involvement, lactate dehydrogenase, serum albumin, D-dimers, oxygen flow, and myoglobin. Based on these parameters we developed a new score (COV-Score). *Conclusions*: Quantitative assessment of lung lesions improves the prediction algorithms compared to the semi-quantitative parameters. The cluster evaluation algorithm increases the non-survivor and overall prediction accuracy.COV-Score represents a viable alternative to current prediction scores, demonstrating improved sensitivity and specificity in predicting mortality at the time of admission.

## 1. Introduction

COVID19 induced by severe acute respiratory syndrome coronavirus 2 (SARS-CoV-2) was first reported in Wuhan City, China, in December 2019 [1], causing approximately 500 million cumulative cases and 6million deaths worldwide by April 2022 [2]. The severe forms of pneumonia are associated with acute hypoxic respiratory failure and high mortality rates, raising significant challenges for the medical community. The current coronavirus evolution models and prognosis scores used in the evaluation and management of severe cases only include qualitative or semi-quantitative evaluations of pulmonary lesions [3,4], which lead to a less accurate prognosis and assessment of pulmonary involvement. Therefore, in the present paper, we aim to present: (1) a new prognosis and monitoring tool with better accuracy than current imaging parameters—an evaluation algorithm using a cluster analysis method that calculates the isolated normal lung volumes, surrounded by inflammatory changes; (2) a quantitative assessment method of lung lesions presented which improves the prediction algorithms for mortality and the management of severe patients upon admission; and (3) a new, more accurate predictive score which includes quantitative radiological parameters.

## 2. Materials and Methods

We performed a retrospective observational cohort study that included 100 adult patients admitted with confirmed severe COVID19 in our department between April 2020 and December 2020. The patients were divided into two groups: the surviving patients were included in group A (76 patients) and the non-survivors in group B (24 patients). The severe form of COVID-19 was defined as at least one of the following criteria: peripheral oxygen saturation (SpO_2_) ≤ 93% in ambient air, respiratory rate (RR) > 30/min, arterial oxygen partial pressure to fractional inspired oxygen ratio (PaO_2_/FiO_2_ ratio) < 300, or lung infiltrates > 50% of lung parenchyma [5]. Inclusion criteria were: age over 18 years, confirmed COVID-19 by RT-PCR (Real-Time—Polymerase Chain Reaction) or rapid antigen testing, chest CT (computer tomography) scan at admission. Exclusion criteria were age under 18 years, pregnant or breastfeeding women, and no chest CT evaluation at admission.

Collected data included demographic and lifestyle parameters (age, sex, smoking status), anthropometric parameters (weight, height, body mass index), clinical parameters (heart rate, systolic and diastolic blood pressure, consciousness status), respiratory function parameters (respiratory rate, SpO_2_ by pulse oximetry and/or SpO_2_ by arterial blood gasses analysis, along with PaO_2_ and PaO_2_/FiO_2_ ratio), complete blood count, coagulation parameters (D-dimers, plasminogen activator inhibitor-1(PAI-1), international normalized ratio (INR)), inflammation markers (C-reactive protein(CRP), interleukin 6 (IL-6), ferritin), biochemical serum parameters (brain natriuretic peptide, troponin I, creatine kinase including MB isoform, myoglobin, serum albumin, lactate dehydrogenase, alanine transaminase, creatinine). Comorbidities included arterial hypertension, diabetes mellitus, coronary artery disease, congestive heart failure (CHF), chronic kidney disease (CKD), obesity, asthma, chronic obstructive pulmonary disorder (COPD), cancer, and any state of immunosuppression. The median Charlson Comorbidity Index (CCI) was calculated for all patients [6]. We assessed the severity of lung involvement by native chest CT examination on a 64-slice CT Somatom Definition As (Siemens, Munich, Germany), using a pitch of 1.2, collimation of 1.2 mm, 3 mm reconstructions with mediastinal (B31f image filter), and a lung window (B80f ultra-sharp image filter). We performed the quantitative lung assessment using SyngoPulmo 3D. After the loading of the CT images into SyngoPulmo 3D, segmentation of lung parenchyma is performed automatically (with possibility of manual adjustment) (Appendix A), excluding the main bronchi and main pulmonary vessels from the pulmonary densitometric analysis (Appendix A); the software performs further the calculation of the lung volume for specific density ranges defined by the user (Appendix A). We calculated the percent of lung involvement based on density ranges, considering alveolar lesions for densities higher than 0 Hounsfield units (HU), mixt lesions (alveolar and interstitial) for densities between 0 and −200 HU, interstitial lesions for densities between −200 and −800 HU, normal parenchyma for densities between −800 and −1000 HU, and emphysema or hyperinflation for densities lower than −1000 HU [7]. We further analyzed the normal lung densities using a cluster analysis method that calculates and displays volumes of connected voxels in a specified density range. For the normal lung densities, we used the following cluster types: cluster1 (C1) between 2 and 10 mm^3^, cluster2 (C2) between 11 and 60 mm^3^, cluster3 (C3) between 61 and 200 mm^3^, and cluster4 (C4) over 201 mm^3^. Logistic regression was performed to assess how various parameters impact mortality in order to establish the best predictive model, and to develop COV-Score, which was further compared to existing scores (MuLBSTA respectively Smart COP). MuLBSTA represents a predictive tool for assessing the mortality risk of viral pneumonia, based on the following parameters: age, lymphocyte count, smoking status, arterial hypertension, the presence of multilobular infiltrates, and the presence of bacterial infection [3]. SmartCOP predicts 30-day mortality in community-acquired pneumonia and establishes which patients will require vasopressor or respiratory support [4]. The required data are systolic blood pressure, multilobar involvement in chest radiography, albumin level, respiratory rate, tachycardia, confusion, low oxygenation, and arterial pH.

For sample size, we considered a confidence level of 95%, a margin of error of 5%, and a population proportion of 6% (representing the prevalence of severe cases in our department). We calculated the minimum sample size using following formula: *n* = [z^2^ × p^ × (1 − p^)]/ε^2^] (*n* = sample size, z = z score, p^ = population proportion, ɛ = margin of error), resulting in a population of 87 patients.

For the statistical analysis, we used the Statistical Package for Social Sciences (SPSS version 25, IBM Corp., Armonk, NY, USA). Patient data are presented as medians and quartiles (Q1, Q3) for continuous variables and as percentages for the categorical variables. We used a Mann-Whitney U-test for continuous variables and Pearson Chi-Square for categorical variables. We evaluated the impact of all parameters on survival rate, mortality rate, and the overall accuracy of prediction using binary logistic regression univariate analyses with the registered data as independent variables and mortality as a dependent variable, followed by multivariable analysis. We used a Pearson correlation to establish which independent variable presented the highest correlation with mortality, which is to be used in the prediction model. We used the ANOVA test to verify the statistical significance, where the *p*-value provided by the Pearson correlation was near 0.05. We used OR to calculate the individual rate of mortality for the independent variables: for OR > 1 the correlation between mortality and independent variable is proportional and the mortality rate is calculated using the formula: (OR − 1) × 100%; for OR < 1 the correlation between mortality and independent variable is inversely proportional and the mortality rate is calculated using the formula: (1 − OR) × 100%. To calculate “double % mortality” we use the following formulas: 1/(OR − 1) for OR > 1 and 1/(1 − OR) for OR < 1. The independent variables were excluded based on the results of the Wald test for the individual parameters obtained during logistic regression analysis; the least significant effect with *p* < 0.2 was removed from the prognostic model. Based on the best logistic model we developed a new prognostic score (COV-Score), to predict mortality in SARS-CoV-2 pneumonia. We performed an ROC curve analysis for every variable in the regression model to identify the cut-off value with optimal specificity and sensitivity; the obtained data were used to create a prognosis score (COV-Score) for the enrolled patients. The statistical significance of logistic regression was estimated by the Omnibus test of model coefficients and verified by the Hosmer-Lemeshow test. We used the Nagelkerke R Square to determine the weights of the dependent variable in the prediction model. A *p*-value lower than 0.05 was considered statistically significant. We compared the COV-Score to existing scores (MuLBSTA respectively Smart COP) using logistic regression and ROC curve analysis. The cutoff point was established by finding the most distant point on the ROC curve from the diagonal (Youden’s J).

This study was performed in line with the principles of the Declaration of Helsinki. Approval was granted by the local Ethical Committee. A waiver for patient consent was given, as this was a retrospective analysis of data already on the medical records and patient-data confidentiality was maintained.

## 3. Results

### 3.1. Clinical, Paraclinical and Laboratory Data

We included 100 adult patients with severe COVID-19 pneumonia, and we divided them into two groups; group A (survivors) included 76 patients with a median age of 57.5 (49.2–65) years. Group B (non-survivors) included 24 patients with a median age of 66 (62–71) years.

Sixty (78.9%) patients from group A and 23 (95.8%) patients in group B presentedcomorbidities (Table 1) with a median CCI score of 1.5 [1; 2] in group A and 3 [2; 6] in group B.

All patients required supplemental oxygen at admission (Table 2), with higher oxygen flow and a lower PaO_2_/FiO_2_ ratio for patients in Group B.

In group B, lower values were detected for the PaO_2_/FiO_2_ ratio, lymphocytes, platelets, and hemoglobin and higher values for neutrophils, BNP, D-dimers, creatine kinase, lactate dehydrogenase, and serum creatinine compared with group A, as shown in Table 3. For the other measured parameters, we found no statistical difference in value distribution.

A higher percentage of lung involvement was found in group B, including alveolar, interstitial, and mixt lesions, with a dominant interstitial component. The distribution of normal lung densities differs between the two groups of patients, with a higher percentage of normal lung densities surrounded by inflammatory changes in group B.

The characterization of lung lesions using semi-quantitative assessment tools (number of pulmonary lobes with pathologic changes) had no statistical significance between Group A and Group B (Table 4).

We performed additional multivariable logistic regression analysis for radiological parameters, with no correlation with the risk of mortality (*p* = 0.166) for qualitative and semi-quantitative independent variables(number of lobes with interstitial lesions, alveolar lesions, and atelectatic lesions). The multivariable logistic regression analysis using quantitative parameters (percent of alveolar lesions, mixt lesions, interstitial lesions, total lung involvement) as independent variables showed statistical significance (*p* = 0.008).

### 3.2. Risk Factors Associated with Higher Mortality

A lower percentage of normal lung densities, PaO_2_/FiO_2_ ratio, lymphocytes, platelets, hemoglobin and serum albumin, and a higher percentage of interstitial lesions, oxygen flow, FiO_2_, Neutrophils/lymphocytes ratio, lactate dehydrogenase, creatine kinase MB, myoglobin, and serum creatinine are associated with higher mortality (Table 5).

The mortality rate doubled for every additional 19.2 years of age, with 10.8 L/min oxygen requirement, 22.2% FiO_2_, 14.9 value of Neu/Ly, 21.3 U/L of CKMB, 200 U/L of LDH, 166.6 µg/L increase in myoglobin. The mortality rate also doubled for every decrease with 100 of PaO_2_/FiO_2_ ratio, 333 (×10^3^/µL) in lymphocytes count, 100 (×10^3^/µL) in platelets count, and 1.06 (g/L) in serum albumin.

A 100% increase in mortality rate was observed for every 15.3% additional interstitial lesions, 20.4% additional total lung involvement, and every 21.2% decrease in normal lung densities.

Evaluating the cluster densities, the mortality rate increasedby 100% for every increase with 1.2% in Cluster 1, 1.07% in Cluster 2, 0.25% in Cluster 3 and for every decrease by 22.7% in Cluster 4.

### 3.3. Regression Model to Evaluate Mortality

The impact of demographic, hematologic, biologic, radiologic, and respiratory variables on survival rate, mortality rate, and the overall accuracy of prediction have been assessed by logistic regression.

From all analyzed parameters using binary logistic regression, Pearson correlation and multivariable logistic analysis we selected the following variables to establish an optimal regression model: age, lymphocytes, PaO_2_/FiO_2_ ratio, percent of lung involvement, lactate dehydrogenase, serum albumin, D-dimers, oxygen flow, and myoglobin (Table 6).

### 3.4. Prognosis Score (COV-Score)

A ROC curve analysis was performed for every variable in the regression model to identify the cut-off value with optimal specificity and sensitivity: age (Appendix A), lymphocytes (Appendix A), PaO_2_/FiO_2_ ratio (Appendix A), percent of lung involvement (Appendix A), lactate dehydrogenase (Appendix A), serum albumin (Appendix A), D-dimers (Appendix A), oxygen flow (Appendix A), and myoglobin (Appendix A).

The obtained data were used to create a prognosis score (COV-Score) for the enrolled patients. We awarded 2 points for “percent of lung involvement” > 60% and 1 point for each of the following parameters: age > 65 years, lymphocytes < 775 (×10^3^/µL), PaO_2_/FiO_2_ ratio < 140, LDH > 450 (U/L), serum albumin < 3.6 g/L, D-dimers > 290 ng/mL, oxygen flow > 14.5 L/min, and myoglobin > 235 µg/L. The points were awarded based on OR of the parameters in the regression model (1 point for each parameter with OR < 1.1). Due to the high impact of radiological parameters in characterizing the mortality (OR ranging from 0.95 to 4.92), we awarded 1 additional point for “total lung involvement”.

We compared the obtained COV-Score using logistic regression with similar scores with radiological evaluation (MuLBSTA and Smart COP)with thebetter overall performance for COV-Score (Table 7).

The description of the variable in the equation for the three scores, OR and CI are provided in Table 8.

Based on the data in Table 8, we can also calculate the probability of death according to COV-Score using the following formula: EXP (Constant + 0.808 × COV-Score)/[1 + EXP (Constant + 0.808 × COV-Score)]. For a COV-Score of 6, we obtain *p* = EXP (−4.963 + 0.808 × 6)/[1 + EXP (−4.963 + 0.808 × 6)] = 0.89/1.89 = 47%.

In Table 9 can be found the death probabilities for all values of COV-Score from 1 to 10.

We performed an additional ROC curve analysis to verify the obtained results by logistic regression and the prediction accuracy for each evolution score. COV-Score (Appendix A) presented a larger AUC (0.884, *p* < 0.001) compared to MuLBSTA (Appendix A), respectively Smart COP (Appendix A), and was able to provide better prediction: for similar specificities (0.75–0.78), compared to MuLBSTA, and Smart COP, we obtained a better sensitivity (0.87 vs. 0.7 and 0.52) respectively for similar sensitivities (0.87–0.9) we obtained a better specificity (0.77 vs. 0.28 and 0.18) (Table 10) (Appendix A).

## 4. Discussion

The analyzed data regarding the radiologic evaluation of patients with severe forms of COVID-19 pneumonia demonstrate a limited value of qualitative and semi-quantitative parameters in establishing an accurate prognosis model. In our study, all patients presented involvement of all five pulmonary lobes in case of interstitial lesions and similar lesion extent in case of alveolar lesions and atelectatic changes, leading to similar distribution of values in both groups A and B and a lack of correlation between the qualitative and semi-quantitative parameters and the risk of death. The data obtained from the quantitative pulmonary evaluation showed an independent correlation with the risk of death for both the differentiated evaluation of pulmonary lesions (alveolar, mixt, and interstitial) and in the global assessment of lung involvement, with statistical significance and good overall prediction accuracy in the prognosis logistic models.

The quantitative lung assessment is obtained using dedicated medical software which allows the evaluation of pulmonary lesions based on mathematical algorithms, considering all voxels from the lung volume included in a density interval defined by the software user, providing an objective result that is quantifiable, reproducible and accurate. We want to emphasize that not all pathologic density voxels are grouped; there are areas with normal density voxels surrounded by areas with pathological voxels, thus in the same pulmonary lobe, we have areas with inflammatory changes alternating with areas with normal lung parenchyma. When normal lung areas are isolated/surrounded by areas with inflammatory changes, there is a high probability that the inflammatory process will rapidly extend to those normal lung areas, therefore, estimating the isolated normal lung volumes will lead to an improvement in the assessment of lung involvement. A larger percent of isolated normal lung areas (C1, C2, and C3) was correlated with poor outcomes in the study group, while a larger percent of normal lung areas connected with similar normal areas and not surrounded by inflammatory changes (C4) is correlated with lower risk of death. The cluster analysis method for normal pulmonary densities represents an important element to consider at patient admission, improving the prediction models.

COV-Score represents a new prognostic score for the outcome of severe SARS-CoV-2 pneumonia, based on predictor variables that are significantly associated with mortality: percent of lung involvement, age, lymphocyte count, PaO_2_/FiO_2_ ratio, LDH, serum albumin, oxygen flow requirement, and myoglobin. This new score introduces an objective evaluation of pneumonia by CT scan, which allows both the differentiation of mild cases from moderate or severe cases and an accurate quantitative assessment of each type of lung involvement. The use of dedicated software as a diagnostic tool allows for an objective lung evaluation, providing a useful tool to initially assess the death risk and adequately monitor the evolution and therapeutic response of the patient. Elderly patients generally have a greater prevalence of positive laboratory tests, such as leukocytosis, lymphopenia, decreased platelet count and anemia, lower albumin, increased D-dimers, and prolonged INR (changes similar to our findings for the patients in Group B), but increased age itself represents an independent risk factor for death of COVID-19 [8], in concordance with results found in our study. Wagner J and al. communicated that lymphocytopenia can be used as an early prognostic factor in determining the clinical evolution and disease severity of ahospitalized COVID-19 patient, with an OR of 3.40 (95% CI 1.06–10.96; *p* = 0.04) inpatients with severe forms of COVID-19 [9]. Although a lower PaO_2_/FiO_2_ ratio and higher O_2_ flow represent independent risk factors associated with the risk of death, the use of high-flow nasal oxygen upon admission in adult patients with COVID-19 may lead to an increase in ventilator-free days and a reduction in ICU length of stay when compared to the early initiation of invasive mechanical ventilation [10]. Elevated LDH can indicate a 44% posterior probability, and non-elevated LDH an 11% posterior probability for poor prognosis and increased mortality (OR 4.22 (95% CI 2.49 to 7.14) [11], therefore, the association of LDH with other parameters in prognostic models is indicated to increase its prognosis value. Serum albumin demonstrated the capacity to bind SARS-CoV-2 spike protein S1 subunit, reducing its interactions with the cell receptors, the patients with reduced or glycated albumin presented a reduced binding [12], which can explain the capacity of albumin to independently predict the mortality in cases of SARS-CoV-2 infection. Patients with COVID-19 and elevated myoglobin levels are at significantly high risk of severe disease and mortality, the presence of elevated myoglobin being more common than elevated cTnI in patients with severe COVID-19 [37.7 (23.3–52.1%) vs. 30.7% (24.7–37.1%)] and preferred as a cardiac biomarker in case of severe patients [13]. In our study, a myoglobinincrease of 166.6 µg/L raised the odds of death by 100%. D-dimer elevations are a frequent laboratory finding in COVID-19 patients requiring hospitalization, representing a good biomarker, indicative of disease severity which could independently predict in-hospital mortality for a value higher than 2.0 µg/mL [14]. A hypercoagulable state in patients with COVID-19 could be explained by the dysfunction of endothelial cells resulting in excess thrombin generation, increased blood viscosity, the activation of the hypoxia-inducible transcription factor-dependent signaling pathway, or the long-term bed rest and invasive treatment in patients with severe forms of pneumonia [14].

While the corrected survival rate was similar to COV-Score (97.4%), the MuLBSTA score had a corrected mortality rate that was significantly lower (17.4% vs. 47.8%), with a lower overall accuracy percentage of 78.8% (vs. 85.9%) and a lower AUC (0.731 vs. 0.884). In our study, all patients with severe COVID-19 pneumonia presented multilobular infiltrates, and therefore a semi-quantitative evaluation has limited use in establishing the prognosis of severe disease patients.

The corrected survival rate of SmartCOP was 100%, slightly higher than COV-Score (97.4%), and the corrected mortality rate was significantly lower (13% vs. 47.8%), with a lower overall accuracy percentage of 79.8% (vs. 85.9%) and a lower AUC (0.639 vs. 0.884). The main cardiovascular risk factor associated with severe SARS-CoV-2 infection is represented by high blood pressure [15], confirmed in our study, in which 22% of the patients presented high systolic blood pressure, and none of them had a value lower than 90mmHg. Patients in Group A (survivors) had a higher respiratory rate (35) than the patients in Group B (non-survivors), who had a rate of 26.5. A third predictor variable—multilobar chest radiography involvement also has no prediction relevance, because, in the case of severe forms of pneumonia, all patients (survivors and nonsurvivors) presented multilobar chest involvement—in this case, a quantitative assessment is mandatory to evaluate the extent of lung involvement. Therefore, we can emphasize that three of the predictor variables of SmartCOPscore don’t fit very well in the profile of SARS-CoV-2 patients with severe pneumonia and cannot be used accurately as predictors: the low systolic blood pressure, high respiratory rate, and multilobar chest radiography involvement.

There are many comorbidities associated with the patient’s risk of developing a severe form of COVID-19 pneumonia or complicating its progression, such as obesity, diabetes mellitus, cardiovascular pathologies, chronic kidney disease, chronic pulmonary disease, chronic hepatitis, oncologic pathologies, and immunity-related diseases [16,17,18,19,20,21,22,23,24]. Therefore, the association of comorbidities represents an important prognostic factor to consider when evaluating the risk of death at admission before performing biological and radiological investigations. In the patients with severe forms of pneumonia (included in our study), it is difficult to obtain relevant anamnestic data or previous medical data to document specific pathologies at the time of admission, and many comorbidities are identified during hospitalization; based on these aspects we did not include any comorbidity in development of the COV-Score.

*Study limitations*: The newly developed COV-Score needs further validation, calibration, and performance studies. MuLBSTA and SmartCOP scores used for comparison with the COV-Score were designed for predicting viral pneumonia and 30-day mortality in community-acquired pneumonia, respectively; the population for the development of these scores might be different from the population used in this study. Patient follow-up was only in hospitals, and data on both 30- and 90-day mortality is lacking. To calculate COV-Score, all patients require CT scans, D-dimers, and myoglobin, which may not be available in all hospitals. Quantitative CT evaluation using clusters of densities is limited by the availability of medical software with the option to calculate specified customizable density clusters. Therefore, for a larger addressability, in the evaluation score (COV-Score), we used the quantitative lung assessment parameter “percent of lung involvement”. Bacterial co-infection may represent a supplementary risk factor, especially in severe cases of patients which may occur after admission to the hospital and increase the mortality rate provided by COV-Score. The prediction model was created based on data from the first waves of the SARS-CoV-2 pandemic, and for applicability with current and future strains, further validation studies may be required.

This article is archived in the preprint server Research Square (https://doi.org/10.21203/rs.3.rs-1594450/v1, accessed on 2 May 2022).

## 5. Conclusions

The uantitative assessment of lung lesions improves the prediction algorithms compared to the semi-quantitative parameters, and hence, we recommend its use as a standard radiological parameter in evaluating COVID-19 pneumonia. The cluster evaluation algorithm of normal lung densities (Cluster 1–Cluster 4) increases the non-survivor and overall prediction accuracy when assessing the prognosis of severe patients. COV-Score represents a viable alternative to current prediction scores (MuLBSTA and SmartCOP), demonstrating a better sensitivity and specificity in predicting a poor outcome at the time of admission. Serum albumin, LDH, lymphocyte count, and degree of lung involvement showed the best correlation with the risk of death.

## Figures and Tables

**Table 1 medicina-58-00945-t001:** Comorbidities in survivors (group A) and non-survivors (group B).

Parameter	Group A *n* (%)	Group B *n* (%)	*p*-Value
Obesity	35 (46.1)	11 (45.8)	>0.05
Diabetes mellitus	13 (17.1)	8 (33.1)	0.03
Arterial hypertension	32 (41.9)	14 (58.3)	>0.05
Congestive heart failure	3 (3.9)	3 (12.5)	>0.05
Chronic kidney disease	3 (3.9)	5 (20.8)	0.03
Chronic obstructive pulmonary disorder	8 (10.5)	4 (16.6)	>0.05
Chronic hepatitis	3 (3.9)	2 (8.3)	>0.05
History of neoplasia	0 (0)	6 (25)	<0.001
Ischemic stroke	2 (2.6)	5 (20.8)	0.01
Dementia	1 (1.3)	1 (4.1)	>0.05
Peptic ulcer	1 (1.3)	0 (0)	>0.05
Other pathologies	25 (32.9)	13 (54.1)	>0.05

**Table 2 medicina-58-00945-t002:** Respiratory function parameters in survivors (group A) and non-survivors (group B).

Parameter	Group A *n* (%)/Median [Q1; Q3]	Group B *n* (%)/Median [Q1; Q3]	*p*-Value
O_2_ by nasal cannula/Venturi mask	17 (22.3)	9 (37.5)	>0.05
O_2_ by non-rebreathing masks	57 (77)	12 (50)	0.02
O_2_ by mechanical ventilation	0 (0)	3 (12.5)	0.003
Oxygen flow L/min	12 [10; 20]	25 [12.5; 30]	0.003
FiO_2_ (%)	60% [50; 60]	60% [48.7; 78.7]	0.01
PaO_2_ (mmHg)	84 [71; 104]	81.5 [61.5; 93.7]	>0.05
PaO_2_/FiO_2_ ratio	163.3 [126.6; 217.9]	110.5 [96.5; 156.9]	0.02
pCO_2_ (mmHg)	37 [34.9; 40]	36 [32; 39.2]	>0.05
RR (breaths/minute)	35 [20.7; 30]	26.5 [20; 28.5]	>0.05

PaO_2_ = arterial O_2_ pressure, FiO_2_ = fractional inspired oxygen, RR = respiratory rate.

**Table 3 medicina-58-00945-t003:** Biologic changes in survivors (group A) and non-survivors (group B).

Parameter	Group A Median [Q1; Q3]	Group B Median [Q1; Q3]	*p*-Value
PaO_2_/FiO_2_ ratio	163.3 [126.6; 217.9]	110.5 [96.5; 156.8]	0.008
Neutrophils (×10^3^/µL)	6200 [4650; 8350]	8750 [5400; 13,675]	0.016
Lymphocytes (×10^3^/µL)	950 [700; 1200]	525 [400; 800]	<0.001
Neutrophils/lymphocytes ratio	6.8 [4.1; 11.4]	14.6 [8.7; 30.8]	<0.001
Platelets (×10^3^/µL)	229.5 [191.7; 318.5]	160 [1110.2; 275.7]	0.004
Hemoglobin (g/dL)	14 [12.8; 14.5]	12.5 [11.3; 13.7]	0.001
BNP (ng/L)	86 [17; 301]	754 [144.7; 2428.7]	0.003
D-dimers (ng/mL)	246 [156; 354]	555 [213; 1141]	0.001
Creatine kinase (U/L)	61.5 [34.2; 127]	191.5 [76; 307.5]	0.004
Myoglobin (µg/L)	120 [83.5; 225.2]	236 [128.3; 332.4]	0.007
Lactate dehydrogenase (U/L)	354.5 [287; 454.2]	570 [457; 680]	0.001
Serum creatinine (mg/dL)	0.9 [0.7; 1.1]	1.4 [1.1; 1.5]	<0.001
Serum albumin (g/L)	3.8 [3.4; 4]	3.35 [3.1; 3.6]	<0.001
IL-6 (pg/mL)	12.6 [2.8; 89.4]	48.6 [5.5; 193.4]	>0.05
Ferritin (ng/mL)	795.2 [289; 1635]	1265.5 [785; 1650]	>0.05
Troponin I (ng/mL)	0.3 [0.3; 0.3]	0.03 [0.03; 0.035]	>0.05
Plasminogen activator inhibitor-1 (ng/mL)	469.2 [289.5; 674.5]	501.5 [211.5; 775.2]	>0.05
Alanine transaminase (U/L)	44.5 [29.2; 86.7]	68 [37; 108]	>0.05
C-reactive protein (mg/L)	63.9 [17.1; 106]	85 [38.2; 146]	>0.05

**Table 4 medicina-58-00945-t004:** CT lung involvement in survivors (group A) and non-survivors (group B) at admission.

Parameter	Group A Median [Q1, Q3]	Group B Median [Q1, Q3]	*p*-Value
alveolar lesions (%)	1.8 [1; 3.2]	3.1 [1.5; 5.1]	0.062
mixt lesions (%)	4.6 [2.5; 7.3]	6.9 [4.1; 12]	0.036
interstitial lesions (%)	39.4 [31.7; 47.8]	49.2 [44.3; 60.1]	0.001
total lung involvement (%)	47.2 [35.9; 63]	64.9 [48.1; 74.1]	0.003
normal lung densities (%)	52.7 [37; 64.1]	35.1 [25.9;51.9]	0.003
Cluster 1 (2–10 mmc) (%)	0.4 [0.2; 0.9]	1.1 [0.3; 1.5]	0.029
Cluster 2 (10–60 mmc) (%)	0.45 [0.2; 0.9]	0.9 [0.3; 1.7]	0.011
Cluster 3 (60–200 mmc) (%)	0.1 [0; 0.3]	0.4 [0.2; 0.6]	0.001
Cluster 4 (over 200 mmc) (%)	51.6 [35.5; 63.7]	33 [20.7; 51.7]	0.002
lobes with interstitial lesions (*n*)	5 [5; 5]	5 [5; 5]	0.931
lobes with alveolar lesions (*n*)	0 [1; 2]	0 [0; 2]	0.306
lobes with atelectatic changes (*n*)	3.5 [2; 5]	2 [2; 4]	0.253

**Table 5 medicina-58-00945-t005:** Risk factors associated with mortality.

Parameter	Pearson Correlation	*p*-Value	OR (95% CI)	Change in Mortality (%)
Age (years)	0.260	0.009	1.052 (1.011; 1.094)	5.2 *
O_2_ flow (L/min)	0.306	0.002	1.092 (1.029; 1.159)	9.2 *
FiO_2_ (%)	0.260	0.009	1.045 (1.009; 1.083)	4.5 *
PaO_2_/FiO_2_ ratio	−0.235	0.02	0.99 (0.982; 0.999)	1 #
Lymphocytes (×10^3^/µL)	−0.360	<0.001	0.997 (0.995; 0.999)	0.3 #
Neutrophils/lymphocytes ratio	0.341	0.001	1.067 (1.023; 1.113)	6.7 *
Platelets (×10^3^/µL)	−0.302	0.002	0.99 (0.984; 0.997)	1 #
Hemoglobin (g/dL)	−0.362	<0.001	0.623 (0.47; 0.83)	38 #
CKMB (U/L)	0.258	0.01	1.047 (1.004; 1.091)	4.7 *
LDH (U/L)	0.371	<0.001	1.005 (1.002; 1.008)	0.5 *
Serum albumin (g/L)	−0.450	<0.001	0.062 (0.012; 0.32)	93.8 #
Myoglobin (µg/L)	0.282	0.013	1.006 (1.001; 1.011)	0.6 *
Interstitial lesions (%)	0.320	0.001	1.065 (1.022; 1.109)	6.5 *
Total lung involvement (%)	0.335	0.001	1.049 (1.018; 12.08)	4.9 *
Normal lung densities (%)	−0.335	0.001	0.953 (0.926; 0.982)	4.7 #
Cluster1 (2–10 mmc) (%)	0.209	0.047	1.798 (1.015; 3.183)	79.8 *
Cluster2 (10–60 mmc) (%)	0.233	0.019	1.936 (1.081; 3.467)	93.6 *
Cluster 3 (60–200 mmc) (%)	0.241	0.016	4.92 (1.262; 19.181)	392 *
Cluster 4 (over 200 mmc) (%)	−0.341	0.001	0.956 (0.93; 0.98)	4.4 #

* for 1 unit increase in the parameter; # for 1 unit decrease in the parameter. OR = odds ratio; PaO_2_ = arterial O_2_ pressure, FiO_2_ = fractional inspired oxygen, CKMB = creatine kinase MB, LDH = lactate dehydrogenase.

**Table 6 medicina-58-00945-t006:** Regression model to evaluate mortality rate in patients with severe COVID-19 pneumonia.

	Omnibus Test of Model Coefficients	Nagelkerke R Square	Hosmer Lemeshow Test	Corrected Survival Rate (%)	Corrected Mortality Rate (%)	Overall Accuracy Prediction (%)
Regression model	<0.001	0.699	0.877	94.3 (97.1 *)	78.6 (85.7 *)	89.8 (93.9 *)

* increase for the corrected survival rate to 97.1%, for the corrected mortality rate to 85.7% and for the overall accuracy prediction to 93.9% if we substitute the parameter “lung involvement” from the regression model with the parameters presented in the cluster analysis (Cluster 1 to Cluster 4).

**Table 7 medicina-58-00945-t007:** Comparative performance for COV-Score, MuLBSTA, and Smart COP.

	Omnibus Test of Model Coefficients	Nagelkerke R Square	Hosmer Lemeshow Test	Corrected Survival Rate (%)	Corrected Mortality Rate (%)	Overall Accuracy Prediction (%)
COV-Score	<0.001	0.5	0.73	97.4	47.8	85.9
MuLBSTA	<0.001	0.18	0.24	97.4	17.4	78.8
Smart COP	<0.01	0.09	0.06	100	13	79.8

**Table 8 medicina-58-00945-t008:** Logistic regression analysis for prediction scores.

Regression Model Type	B	S.E.	Wald	*p*	OR	95% CI for OR	
						Lower	Upper
COV-Score Constant	0.808	0.182	19.749	<0.001	2.243	1.571	3.203
	−4.963	0.983	25.490	<0.001	0.007		
MuLBSTA Constant	0.370	0.116	10.265	0.001	1.448	1.155	1.817
	−4.244	1.037	16.748	<0.001	0.014		
Smart COP Constant	0.430	1.171	6.320	0.012	1.537	1.099	2.150
	−3.498	0.978	12.791	<0.001	0.03		

B = unstandardized coefficient, S.E. = standard error, OR = odds ratio, CI = confidence interval.

**Table 9 medicina-58-00945-t009:** Probability of death for COV-Score.

COV-Score Value	0	1	2	3	4	5	6	7	8	9	10
Probability of death (%)	0	1.5	3.3	7.3	15	28.4	47	66.6	81.7	90.1	95.7
Observed frequency of death (%)	0	0	0	9.1	11.1	30	40	80	100	100	100

**Table 10 medicina-58-00945-t010:** ROC curve analysis for prediction scores.

Regression Model Type	AUC	Std Error	*p*-Value	Cut Off Value 1	Se	Sp	Cut Off Value 2	Se	Sp
COV-Score	0.884	0.036	<0.001	4.5	0.87	0.77	4.5	0.87	0.77
MuLBSTA	0.731	0.061	0.001	8.5	0.7	0.75	6	0,9	0.28
Smart COP	0.639	0.077	0.045	5.5	0.52	0.78	3.5	0.87	0.18

AUC = area under curve, Std error = standard error, Se = sensitivity, Sp = specificity.

## Data Availability

The data presented in this study are available on request from the corresponding author.

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
