# Peer review of "Mortality Predictors in Severe SARS-CoV-2 Infection"

_medicina, 2022, doi:10.3390/medicina58070945_

Round 1
Reviewer 1 Report
The study is about the development of a new prediction score to determine the risk of mortality in patients with SARS-CoV-2 pneumonia using the quantitative radiological parameter as one of the predictors. There were 100 patients participating in the study. Of these, there were 24 cases of mortality. The risk factors for mortality included a lower percentage of normal lung densities, low PF ratio, low lymphocyte count, anemia, thrombocytopenia and hypoalbuminemia. While, a higher percentage of the interstitial lesion, use of high oxygen flow rate, and FiO2, an increase in the neutrophil-to-lymphocyte ratio, high level of lactate dehydrogenase (LDH), creatine kinase MB, myoglobin and serum creatinine were associated with greater mortality.
Moreover, the authors developed a new prediction score name COV-Score from the regression analysis. The predictors involved in the model included age, lymphocyte, PF ratio, % of lung involvement, LDH, serum albumin, D-dimer, oxygen flow rate, as well as myoglobin. In addition, the authors claimed that their COV-Score was better than the previous score types, including MuLBSTA and SmartCOP. As COV-Score composes of quantitative measure of lung involvement by SARS-CoV-2 by CT scan. While others only utilized semi-quantitative measures of multilobar pulmonary infiltration as their predictor.
This study provided a more accurate prediction score for determining mortality in severe COVID-19 pneumonia. However, there are some considerations before proceeding for publication including:
Abstract
- Please provide the time point of performing the CT scan in the abstract. It supposes to be a CT scan at hospitalization.
- Please check for grammatical errors and some writing mistakes for example
- Line 12 “The objective of this study is to present…”
- Line 20 “We found a low percentage of normal lung densities,….., and serum albumin were associated with a higher mortality.”
- Line 30 A space after a period, before COV-Score.
- Line 31 Please change the word “poor outcome” to “mortality”, owing to a more specific term.
Materials and Methods
- Please provide a full annotation for the word “IMC” on line 67.
- Please check for relevant data. Some data were mentioned in this section but there were not shown in the table, including PAI-1, IL-6, ferritin, troponin I, and ALT.
- It would be worth providing some pictures that represent how the authors quantitate the CT scan imaging.
- Please provide which time point defines mortality rate (hospitalization, 30-day, etc.).
- Please provide the sample size calculation that is needed to answer the hypothesis of the study.
- Please provide more details in regard to statistical analyses for searching risks of mortality. Did the authors perform a screening of significant variables using univariable logistic regression analysis? Then, significant variables were determined using multivariable logistic regression analysis. In addition, if so, please provide the criteria that were used as a selection criterion.
- For a prediction score, some stats still need to be performed such as model validation, calibration, and performance according to the TRIPOD statement. More information about how the authors selected the predictors and assigned the score should be mentioned in the stat section as well.
- Line 105 Beside percentage, please provide the meaning of the number in parenthesis in Table 2. As well, mention the meaning in a footnote at the bottom of the table.
- Please give more explanation why the ANOVA test was needed for statistical significance at Line 108. Generally, ANOVA is used for comparing more than 2 means. In addition, using Pearson correlation to identify the relationship between mortality and predictors is accepted. However, the authors should provide the criteria for variable selection from this analysis as well (maybe use the degree of correlation r>0.50 as a criterion).
- Please provide the information about informed consent.
Results
- Please summarize the data in paragraph 3.1 in a table form and please provide their corresponding p-value as well.
- Header of Table 2 “Parameter” unit is a percentage, not median (IQR).
- It will be clearer to change the title of Table 3 from “poor outcome” to “mortality”.
- Please change “ExpB (95%CI for ExpB)” at the header of Table 3 to “OR (95% CI)”.
- Please provide how to calculate “% mortality” (Table 3) in a stat section. In addition, please provide how to calculate “double % mortality” (line 169 and so on) as well.
- Please provide a reason for analyzing using Pearson correlation. Is Pearson correlation involved in a predictor selection? If so, please provide the selection criteria. If not, please provide the method for selecting (Table 3).
- For ease of reading, please organize Table 3 variables in a corresponding order to the material and method section. For instance, age should come first and radiological information should come later.
- Please delete “survival” and “rate” for subsection 3.3 to be “Regression model to evaluate mortality” and at the title of Table 4 as well.
- Line 184: Please provide the parameters of selection criteria.
- Table 4: Besides the performance of COV-Score, please provide the performance of SmartCOP and MuLBSTA as well. Also, please give a footnote for the number in the parenthesis. Are they a standard error?
- Please show the ROC of every variable (line 194). It may present as a supplement material if needed. In addition, please provide sufficient detail on where the assigned points are from.
- Line 203: It is not clear what the value of Nagelkerke R Sq 0.5 vs 0.18 respectively 0.09. Please provide more details about what 0.09 is. In addition, this also includes 0.06 at line 204 and 79.8% at line 207.
- The Nagelkerke R Sq and HL test for COV-Score in Table 4 was 0.699 and 0.877 respectively, which are not matched to the text provided on lines 203-204.
- Besides providing the probability of death, the observed frequency of death should also be in Table 6.
- Line 221-223: Please provide the p-value for a larger AuROC of COV-Score over other scores.
- Please provide a ROC graph for all scores corresponding to Table 7. In addition, please mention how the authors find the cutoff point in the stat section.
Discussion
- Paragraph 1 of the discussion will be more accurate if the authors perform sufficient and suitable multivariable logistic regression analysis. Only crude evidence from univariable analyses presented in Table 2 is not robust enough to conclude this paragraph. This one also applies to paragraph 2 (Line 248-252) as well.
- Please present why the risk factors for mortality were somewhat different from the COV-Score. In particular, age and lymphocyte count are in the score but are not in the risk for mortality.
- Please be aware that aging, Line 267-269, is not the author's hypothesis and had not been mentioned in the result section before. Therefore, please consider rewriting this sentence.
- Please provide a paragraph discussing the limitations of the study. Therefore, lines 253-257 should move to this section. Moreover, the authors should mention that MuLBSTA and SmartCOP were firstly designed for predicting viral pneumonia and 30-day mortality in CAP, respectively. The population for the development of these scores might be different from the author's study.
- Please check grammar for Line 278-279.
- Line 289-290 are not in the earlier results section. Please consider this sentence.
- Please check the corrected survival rate (97.4%) on lines 298 and 308. It does not match the table.
- Bacterial co-infection is not mentioned earlier. Therefore, it should be deleted or may be printed as one limitation of the study.
- Please check the number provided for respiratory rate (Line 312-313). It is contradicted by the result section (Line 139-140).
- Line 322-328 are about comorbidities that relate to mortality. The authors could mention at the end of the paragraph that COV-Score does not include any comorbidity at all.
Author Response
Thank you for giving us the opportunity to submit a revised draft of the manuscript “Mortality predictors in severe SARS-CoV-2 infection” for publication in “Medicina”. We appreciate the time and effort that you dedicated to providing feedback on our manuscript and we are grateful for the insightful comments on and valuable improvements to our paper. We have incorporated the suggestions in the manuscript. Please find the attached file with our point-by-point response to your comments and concerns.
Abstract
Q1. Please provide the time point of performing the CT scan in the abstract. It supposes to be a CT scan at hospitalization.
A1. We introduced following sentence: “All patients were evaluated by CT scan at admission in hospital.”
Q2. Please check for grammatical errors and some writing mistakes for example
- Line 12 “The objective of this study is to present…”
- Line 20 “We found a low percentage of normal lung densities,….., and serum albumin were associated with a higher mortality.”
- Line 30 A space after a period, before COV-Score.
- Line 31 Please change the word “poor outcome” to “mortality”, owing to a more specific term.
A2. We corrected the mentioned grammatical errors and we addressed to an english editing service to correct additional errors.
Materials and Methods
Q3. Please provide a full annotation for the word “IMC” on line 67.
A3: We replaced “IMC” with body mass index.
Q4. Please check for relevant data. Some data were mentioned in this section but there were not shown in the table, including PAI-1, IL-6, ferritin, troponin I, and ALT.
A4: We showed in table only the data with p<0.05. We included in Table 1 the suggested parameters (IL6, ferritin, troponin I, PAI-1, ALT and CRP)
Q5. It would be worth providing some pictures that represent how the authors quantitate the CT scan imaging.
A5. Following sentence was added in Materials and Metods: “After the loading of the CT images into SyngoPulmo 3D, segmentation of lung parenchyma is performed automatically (with possibility of manual adjustment) (Figure S1), excluding the main bronchi and main pulmonary vessels from the pulmonary densitometric analysis (Figure S2); the software performes further the calculation of the lung volume for specific density ranges defined by the user (Figure S3)”. The requested pictures were provided as supplementary material.
Q6. Please provide which time point defines mortality rate (hospitalization, 30-day, etc.).
A6.We quantified the mortality rate during hospitalization.
Q7.Please provide the sample size calculation that is needed to answer the hypothesis of the study.
A7.For sample size we considered a confidence level of 95%, a margin of error of 5% and a population proportion of 6% (representing the prevalence of severe cases in our department). We calculated the minimum sample size using following formula: n=[z2 x p̂ x (1- p̂)]/ɛ2] (n=sample size, z = z score, p̂ = population proportion, ɛ=margin of error), resulting a population of 87 patients.
Q8. Please provide more details in regard to statistical analyses for searching risks of mortality. Did the authors perform a screening of significant variables using univariable logistic regression analysis? Then, significant variables were determined using multivariable logistic regression analysis. In addition, if so, please provide the criteria that were used as a selection criterion.
A8.To search the risk factors of mortality, we first performed binary logistic regression univariate analyses with the registered data as independent variables and mortality as dependent variable, followed by multivariable analysis.
We used Pearson correlation to establish which independent variable presented the highest correlation with mortality, to be used in the prediction model.
Q9.For a prediction score, some stats still need to be performed such as model validation, calibration, and performance according to the TRIPOD statement. More information about how the authors selected the predictors and assigned the score should be mentioned in the stat section as well.
A9.We introduced a subsection of study limitation where we mentioned that “The new developed COV-Score needs further validation, calibration and performance studies”
We introduced more information about the selection of predictors and score assignment in the stat section.
Q10.Line 105 Beside percentage, please provide the meaning of the number in parenthesis in Table 2. As well, mention the meaning in a footnote at the bottom of the table.
A10: The meaning of the number in parenthesis in Table 2 is Q1 and Q3 (first and third quartile)
Q11. Please give more explanation why the ANOVA test was needed for statistical significance at Line 108. Generally, ANOVA is used for comparing more than 2 means. In addition, using Pearson correlation to identify the relationship between mortality and predictors is accepted. However, the authors should provide the criteria for variable selection from this analysis as well (maybe use the degree of correlation r>0.50 as a criterion).
A11: We used ANOVA to verify the statistic significance where p-value provided by Pearson correlation was near 0.05
Q12. Please provide the information about informed consent.
A12:We introduced additional information regarding the informed consent: “A waiver for patient consent was given as this was a retrospective analysis of data already on the medical records and patient-data confidentiality was maintained.”
Results
Q13. Please summarize the data in paragraph 3.1 in a table form and please provide their corresponding p-value as well.
A13:We introduced the data in paragraph 3.1 in two tables (Table 1 and Table 2) and provided the corresponding p-value for each parameter.
Q14. Header of Table 2 “Parameter” unit is a percentage, not median (IQR).
A14:We modified the header of Table 2.
Q15. It will be clearer to change the title of Table 3 from “poor outcome” to “mortality”.
A15:We changed the title in : “Risk factors associated with mortality”
Q16. Please change “ExpB (95%CI for ExpB)” at the header of Table 3 to “OR (95% CI)”.
A16: We replaced ExpB (95%CI for ExpB) with OR (95% CI)
Q17. Please provide how to calculate “% mortality” (Table 3) in a stat section. In addition, please provide how to calculate “double % mortality” (line 169 and so on) as well.
A17.We introduced in stat section: “We used OR to calculate the individual rate of mortality for the independent variables: for OR>1 the correlation between mortality and independent variable is proportional and the rate of mortality is calculated using as formula: (OR-1) x100%; for OR<1 the correlation between mortality and independent variable is inversely proportional and the risk of mortality is calculated using as furmula: (1-OR)x100%.”
To calculate double % mortality rate we use following formula: 1/(OR-1) for OR>1 and 1/(1-OR) for OR<1
Q18. Please provide a reason for analyzing using Pearson correlation. Is Pearson correlation involved in a predictor selection? If so, please provide the selection criteria. If not, please provide the method for selecting (Table 3).
A18:We used Pearson correlation to establish which independent variable presented the highest correlation with mortality, to be used in the prediction model. We selected the variables with the highest correlation and further analyze them by multivariable logistic regression analysis in order to obtain the best regression model.
Q19. For ease of reading, please organize Table 3 variables in a corresponding order to the material and method section. For instance, age should come first and radiological information should come later.
A19: We organized Table 3 variables as suggested.
Q20. Please delete “survival” and “rate” for subsection 3.3 to be “Regression model to evaluate mortality” and at the title of Table 4 as well.
A20:We performed the suggested changes for subsection 3.3 and Table 4
Q21. Line 184: Please provide the parameters of selection criteria.
A21:We rephrased the sentence as follows: “From all analyzed parameters using binary logistic regression, Pearson correlation and multivariable logistic analysis we selected following variables to establish an optim regression model: age, lymphocytes, PaO2/FiO2 ratio, percent of lung involvement, lactate dehydrogenase, serum albumin, D-dimers, oxygen flow, and myoglobin”
Q22. Table 4: Besides the performance of COV-Score, please provide the performance of SmartCOP and MuLBSTA as well. Also, please give a footnote for the number in the parenthesis. Are they a standard error?
A22:In Table 4 is presented the multivariable regression model for age, lymphocytes, PaO2/FiO2 ratio, percent of lung involvement, lactate dehydrogenase, serum albumin, D-dimers, oxygen flow, and myoglobin.
The performance of COV-Score compared to SmartCOP and MuLBSTA are provided in 3.4
A footnote was added to explain the number in parenthesis: “* increase for the corrected survival rate to 97,1%, for the corrected mortality rate to 85,7% and for the overall accuracy prediction to 93,9% if we substitute the parameter “lung involvement” from the regression model with the parameters presented in the cluster analysis (Cluster 1 to Cluster 4)”
Q23. Please show the ROC of every variable (line 194). It may present as a supplement material if needed. In addition, please provide sufficient detail on where the assigned points are from.
A23: ROC curve for every variable was provided as supplementary material (Figure S4-Figure S12). The points were awarded based on OR of the parameters found in “Table 3. Risk factors associated with mortality” (1 point for each parameter with OR<1,1). Due to the high impact of radiological parameters in characterizing the mortality (OR ranging from 0,95 to 4,92), we awarded 1 additional point to “total lung involvement”.
Q24. Line 203: It is not clear what the value of Nagelkerke R Sq 0.5 vs 0.18 respectively 0.09. Please provide more details about what 0.09 is. In addition, this also includes 0.06 at line 204 and 79.8% at line 207.
A24: The three values in parenthesis represents the Nagelkerke R Square, p-value for Hosmer-Lemeshow test, prediction of corrected mortality rate, and overall accuracy prediction for OV-Score vs MuLBSTA vs Smart COP. For clarification, we presented the mentioned data in a new table.
Q25. The Nagelkerke R Sq and HL test for COV-Score in Table 4 was 0.699 and 0.877 respectively, which are not matched to the text provided on lines 203-204.
A25: The Nagelkerke R Sq and HL test for COV-Score in Table 4 are for the multivariable logistic regression performed on age, lymphocytes, PaO2/FiO2 ratio, percent of lung involvement, lactate dehydrogenase, serum albumin, D-dimers, oxygen flow, and myoglobin.
The other values provided on lines 203-204 are for the binary logistic regression performed on the actual COV-Score.
Q26. Besides providing the probability of death, the observed frequency of death should also be in Table 6.
A26: We introduced in the table the observed frequency of death
Q27. Line 221-223: Please provide the p-value for a larger AuROC of COV-Score over other scores.
A27: we introduced in the text the requested modifications “larger AUC (0,884, p<0,001)”
Q28. Please provide a ROC graph for all scores corresponding to Table 7. In addition, please mention how the authors find the cutoff point in the stat section.
A28: ROC curve for every variable was provided as supplementary material (Figure S13, Figure S14, Figure S15). The cutoff point was established by searching the most distant point on the ROC curve from the diagonal.
Discussion
Q29. Paragraph 1 of the discussion will be more accurate if the authors perform sufficient and suitable multivariable logistic regression analysis. Only crude evidence from univariable analyses presented in Table 2 is not robust enough to conclude this paragraph. This one also applies to paragraph 2 (Line 248-252) as well.
A29: We added in Results the results obtained from multivariable logistic regression analysis for both semi-quantitative parameters and quantitative parameters:
“We performed additional multivariable logistic regression analysis for radiological parameters, with no correlation with the risk of mortality (p=0,166) for qualitative and semi-quantitative independent variables (number of lobes with interstitial lesions, alveolar lesions and atelectatic lesions). The multivariable logistic regression analysis using quantitative parameters (percent of alveolar lesions, mixt lesions, interstitial lesions, total lung involvement) as independent variables showed statistic significance (p=0,008)”
Q30. Please present why the risk factors for mortality were somewhat different from the COV-Score. In particular, age and lymphocyte count are in the score but are not in the risk for mortality.
A30: Age and lymphocyte count are present in the table with risk factors for mortality. We rearranged parameters for a better visibility as suggested at Q17.
Q31. Please be aware that aging, Line 267-269, is not the author's hypothesis and had not been mentioned in the result section before. Therefore, please consider rewriting this sentence.
A31: We rephrased the sentence: “Elderly patients have generally a greater prevalence of positive laboratory tests, such as leukocytosis, lymphopenia, decreased platelet count and anemia, lower albumin, increased D-dimers, and prolonged INR (changes similar to our findings for the patients in Group B), but old age itself represents an independent risk factor for death of COVID-19 [8], in concordance with results found in our study.”
Q32. Please provide a paragraph discussing the limitations of the study. Therefore, lines 253-257 should move to this section. Moreover, the authors should mention that MuLBSTA and SmartCOP were firstly designed for predicting viral pneumonia and 30-day mortality in CAP, respectively. The population for the development of these scores might be different from the author's study.
A32: We introduced a subsection with the limitations of the study:
Study limitations: The new developed COV-Score needs further validation, calibration and performance studies. MuLBSTA and SmartCOP scores, used for comparison with COV-Score were designed for predicting viral pneumonia and 30-day mortality in community acquired pneumonia, respectively; the population for the development of these scores might be different from population used in this study. Patient follow-up was only in hospital and data on 30 and 90 day mortality are lacking. To calculate COV-Score all patients require CT scan, D-dimers and myoglobin which may not be available in all hospitals. Quantitative CT evaluation using clusters of densities is limited by the availability of medical software with the option to calculate specified customizable density clusters. Therefore, for a larger addressability, in the evaluation score (COV-Score), we used as quantitative lung assessment parameter “percent of lung involvement”. Bacterial co-infection may represent a supplementary risk factor, especially in severe cases of patients which may occur after admission in hospital and increase the mortality rate provided by COV-Score.
Q33. Please check grammar for Line 278-279.
A33:We rephrased the sentence.
Q34. Line 289-290 are not in the earlier results section. Please consider this sentence.
A34: The information regarding myoglobin can be found in Results section 3.2. Risk factors associated with higher mortality: “The mortality rate doubled for every additional 19,2 years of age, 10,8 L/min oxygen requirement, 22,2% FiO2, 14,9 value of Neu/Ly, 21,3 U/L of CKMB, 200 U/L of LDH, 166,6 µg/L increase in myoglobin. “
The information regarding D-Dimers are from reference 14 (Zhang L, Yan X, Fan Q, et al. D-dimer levels on admission to predict in-hospital mortality in patients with Covid-19. J Thromb Haemost. 2020;18:1324-9. DOI: 10.1111/jth.14859)
“D-dimer elevations are a frequent laboratory finding in COVID-19 patients requiring hospitalization, representing a good biomarker indicative of disease severity which could independently predict in-hospital mortality for a value higher than 2.0µg/mL.”
Q35. Please check the corrected survival rate (97.4%) on lines 298 and 308. It does not match the table.
A35: We introduced in Results section 3.4. Prognosis score (COV-Score) a comparative performance for COV-Score, MuLBSTA and Smart COP where the corrected survival rate are presented.
Q36. Bacterial co-infection is not mentioned earlier. Therefore, it should be deleted or may be printed as one limitation of the study.
A36: We added in the section Study limitations: “Bacterial co-infection may represent a supplementary risk factor, especially in severe cases of patients which may occur after admission in hospital and increase the mortality rate provided by COV-Score. “
Q37. Please check the number provided for respiratory rate (Line 312-313). It is contradicted by the result section (Line 139-140).
A37: In results, patients in Group A presented a RR of 35 [20,7-30] breaths /minute vs 26,5 [20; 28,5] breaths/minute for patients in Group B.
Q38. Line 322-328 are about comorbidities that relate to mortality. The authors could mention at the end of the paragraph that COV-Score does not include any comorbidity at all.
A38: We added following sentence at the end of the paragraph: “In patients with severe forms of pneumonia (included in our study) is difficult to obtain at admission relevant anamnestic data or previous medical data to document specific pathologies, and many comorbidities are identified during hospitalization; based on these aspects we did not include any comorbidity in development of COV-Score. “

Reviewer 2 Report
This is an interesting study on COVID-19 patients that introduces a new score for predicting mortality in patients admitted between April 2020 and December 2020. I have some concerns, that can be found below:
1. Line 12: ‘The objective of this paper to present’ à Something is missing: ‘was’ to present?
2. Line 21-24: I find it difficult to read and understand this sentence. Perhaps it needs rephrasing
3. English needs correction
4. Line 49: Change ‘ad’ to ‘at’
5. Line 65: What is IMC? Please define all abbreviations when first used
6. The two scores that were used for comparison with the newly developed score in this manuscript were not developed for COVID-19. They are used for pneumonia and were developed years before the pandemic. Thus, it is questionable why they were chosen for this comparison. 4C Mortality score for COVID-19, on the other hand, is used specifically for COVID-19
7. Line 129: Please remove the references from the results section. You could introduce and define the term CCI in the methods section and use the reference there
8. Line 118-141: All the information provided here is important but difficult to follow in the text. Also, statistical comparison should be performed between those who lived and those who survived. A table (maybe supplementary table, since there are already many tables in the manuscript) could present these data in a more friendly way for the reader
9. Table 3: I guess that ExpB is the OR. If yes, please replace it
10. Table 3: Please add all necessary abbreviations at the footnote (LDH, CKMB, FiO2, PaO2, OR)
11. Table 3: At the end of the 3rd line of the results, there is a ‘%’ that is not anywhere in that column. Is that necessary?
12. Table 3: The title says ‘poor outcome’ Do you mean mortality?
13. Results 3.3: Use of myoglobin is questionable and may make this score hard to use. Troponin has a more widespread use. Why didn’t you choose this parameter instead?
14. Table 5: Please define all abbreviations in a footnote
15. Table 5: I guess this is a table as given from SPSS. If yes, I suppose that sig could be replaced by p, and Exp(B) by OR
16. Table 7: Please define all abbreviations in a footnote
17. This manuscript lacks a ‘limitations’ subsection right before the conclusions section, at the end of the discussion section. I feel that the most important limitations are the following: First of all, all these data are from the first, the second and, partially, the third pandemic wave, involving, presumably, the wild type and the alpha, beta and delta variants. Thus, it is highly questionable if all these data and the newly developed score are applicable today with the omicron variant and the immunity provided by vaccination. Secondly, I am afraid that the comparators of the new score in this study are not quite supposed to assess mortality in COVID-19. Probably, 4C score is more likely to be used in most hospitals. Thirdly, use of the new score has limitations, since it requires CT in all patients, and myoglobin – a test not that widely used in COVID-19
Author Response
Thank you for giving us the opportunity to submit a revised draft of the manuscript “Mortality predictors in severe SARS-CoV-2 infection” for publication in “Medicina”. We appreciate the time and effort that you dedicated to providing feedback on our manuscript and we are grateful for the insightful comments on and valuable improvements to our paper. We have incorporated the suggestions in the manuscript. Please find the attached file with our point-by-point response to your comments and concerns.
Q1: Line 12: ‘The objective of this paper to present’ à Something is missing: ‘was’ to present?
A1: We rephrased the sentence according to suggestion: “The objective of this paper is to present the importance of quantitative evaluation of radiological changes in SARS-CoV-2 pneumonia, including an alternative way to evaluate the lung involvement using normal density clusters.”
Q2: Line 21-24: I find it difficult to read and understand this sentence. Perhaps it needs rephrasing
A2: We rephrased the sentence according to suggestion: “We found a low percentage of normal lung densities, PaO2/FiO2 ratio, lymphocytes, platelets, hemoglobin and serum albumin associated with higher mortality; a high percentage of interstitial lesions, oxygen flow, FiO2, Neutrophils/lymphocytes ratio, lactate dehydrogenase, creatine kinase MB, myoglobin, and serum creatinine were associated with higher mortality”
Q3: English needs correction
A3: We addressed to an english editing service.
Q4:Line 49: Change ‘ad’ to ‘at’
A4: We rephrased the sentence according to suggestion: “management of severe patients at admission”
Q5: Line 65: What is IMC? Please define all abbreviations when first used
A5: We replaced IMC with “body mass index”
Q6:The two scores that were used for comparison with the newly developed score in this manuscript were not developed for COVID-19. They are used for pneumonia and were developed years before the pandemic. Thus, it is questionable why they were chosen for this comparison. 4C Mortality score for COVID-19, on the other hand, is used specifically for COVID-19
A6: The two score choosed for comparison (Smart COP and MuLBSTA) are used in medical practice to estimate the prognosis of patients with severe forms of pneumonia, including COVID-19 pneumonia and are using similar categories of parameters (demographic, laboratory and imaging parameters). We compared “COV-Score” with Smart COP and MuLBSTA in order to demonstrate that “COV Score” represents a viable alternative to current prediction scores, with a better sensitivity and specificity in predicting mortality at the time of admission in patients with SARS-CoV-2 infection.
Q7:Line 129: Please remove the references from the results section. You could introduce and define the term CCI in the methods section and use the reference there
A7: We introduced the term CCI in the methods section and removed the reference from the results section.
Q8: Line 118-141: All the information provided here is important but difficult to follow in the text. Also, statistical comparison should be performed between those who lived and those who survived. A table (maybe supplementary table, since there are already many tables in the manuscript) could present these data in a more friendly way for the reader
A8: We presented the comorbidities for both groups (survivors and non-survivors) in “Table 1. Comorbidities in survivors (group A) and non-survivors (group B)”.
We presented the respiratory parameters in “Table 2. Respiratory function parameters in survivors (group A) and non-survivors (group B)”.
We used binary logistic regression to evaluate the statistic significance between the two groups in both tables and we introduced also the obtained data in the text of the article.
Q9: Table 3: I guess that ExpB is the OR. If yes, please replace it
A9: We replaced ExpB with OR
Q10: Table 3: Please add all necessary abbreviations at the footnote (LDH, CKMB, FiO2, PaO2, OR)
A10: We added all necessary abbreviations at the table footnote: “OR= odds ratio; PaO2= arterial O2 pressure, FiO2= fractional inspired oxygen, CKMB= creatine kinase MB, LDH= lactate dehydrogenase”
Q11: Table 3: At the end of the 3rd line of the results, there is a ‘%’ that is not anywhere in that column. Is that necessary?
A11: We removed the “%” from the parameter
Q12:Table 3: The title says ‘poor outcome’ Do you mean mortality?
A12: Yes, we used the term “poor outcome” instead of mortality in Table 3 title. We reformulated the title: “Table 3. Risk factors associated with mortality”
Q13:Results 3.3: Use of myoglobin is questionable and may make this score hard to use. Troponin has a more widespread use. Why didn’t you choose this parameter instead?
A13: We used both binary logistic regression and Pearson correlation to evaluate the variation of all measured parameters and their association with mortality and we found no statistic significance for troponin.
Q14: Table 5: Please define all abbreviations in a footnote
A14: We added all necessary abbreviations at the table footnote:
“B=unstandardized coefficient, SE=standard error, OR=odds ratio, CI= confidence interval”
Q15:Table 5: I guess this is a table as given from SPSS. If yes, I suppose that sig could be replaced by p, and Exp(B) by OR
A15: We replaced sig with p and Exp(B) with OR
Q16: Table 7: Please define all abbreviations in a footnote
A16: We added all necessary abbreviations at the table footnote: “AUC=area under curve, Std error=standard error, Se= sensitivity, Sp= specificity ”
Q17: This manuscript lacks a ‘limitations’ subsection right before the conclusions section, at the end of the discussion section. I feel that the most important limitations are the following: First of all, all these data are from the first, the second and, partially, the third pandemic wave, involving, presumably, the wild type and the alpha, beta and delta variants. Thus, it is highly questionable if all these data and the newly developed score are applicable today with the omicron variant and the immunity provided by vaccination. Secondly, I am afraid that the comparators of the new score in this study are not quite supposed to assess mortality in COVID-19. Probably, 4C score is more likely to be used in most hospitals. Thirdly, use of the new score has limitations, since it requires CT in all patients, and myoglobin – a test not that widely used in COVID-19
A17: We included a subsection of study limitation: “Study limitations: The new developed COV-Score needs further validation, calibration and performance studies. MuLBSTA and SmartCOP scores, used for comparison with COV-Score were designed for predicting viral pneumonia and 30-day mortality in community acquired pneumonia, respectively; the population for the development of these scores might be different from population used in this study. Patient follow-up was only in hospital and data on 30 and 90 day mortality are lacking. To calculate COV-Score all patients require CT scan, D-dimers and myoglobin which may not be available in all hospitals. Quantitative CT evaluation using clusters of densities is limited by the availability of medical software with the option to calculate specified customizable density clusters. Therefore, for a larger addressability, in the evaluation score (COV-Score), we used as quantitative lung assessment parameter “percent of lung involvement”. Bacterial co-infection may represent a supplementary risk factor, especially in severe cases of patients which may occur after admission in hospital and increase the mortality rate provided by COV-Score.”

Round 2
Reviewer 1 Report
Most suggestions have been edited with suitable explanations.
However, I am not familiar with some sort of stats that were used in the manuscript. It would be great if the authors provide some references for them as well. In particular, how to select predictors, how to assign the score, and how to find the cutoff point.
These are what I have usually found for a prediction score development.
1) The backward elimination (or other methods) with a p-value < 0.10-0.20 is often used as a criterion for predictors selection.
2) The prediction score for each independent variable is usually created by calculating the multivariable logistic regression coefficients divided by the lowest coefficients of the model and rounded to the nearest integer.
3) The cutoff point is usually calculated by Youden or Liu's method.
Lastly, I would recommend showing the ROC graph combining all three models including COV-Score, MuLBSTA, and SmartCOP in the same picture. This graph will represent how your COV-Score discriminates against patients who are at high risk for mortality better than the other models.
Author Response
We appreciate your time and work in reviewing our paper and providing valuable and insightful comments, leading to improvements in the current version. The authors have carefully considered the comments and tried our best to address every one of them. We hope the manuscript after the requested revisions meet your high standards.
Q1.The backward elimination (or other methods) with a p-value < 0.10-0.20 is often used as a criterion for predictors selection.
A1. Following paragraph has been added to “Materials and Methods” in statistical analysis subsection : “The independent variables were excluded based on the results of the Wald test for the individual parameters obtained during logistic regression analysis; the least significant effect with p<0.2 was removed from the prognostic model.”
Q2. The prediction score for each independent variable is usually created by calculating the multivariable logistic regression coefficients divided by the lowest coefficients of the model and rounded to the nearest integer.
A2. We explained prediction score for each independent variable in subsection 3.4 Prognosis score (COV-Score):
The points were awarded based on OR of the parameters in the regression model (1 point for each parameter with OR<1.1). Due to the high impact of radiological parameters in characterizing the mortality (OR ranging from 0.95 to 4.92), we awarded 1 additional point to “total lung involvement”.
Q3. The cutoff point is usually calculated by Youden or Liu's method.
A3. The maximum distance between ROC curve and the diagonal line is represented by J (the Youden index).
We added the information in the statistical section: “The cutoff point was established by finding the most distant point on the ROC curve from the diagonal (Youden’s J).”
Q4. Lastly, I would recommend showing the ROC graph combining all three models including COV-Score, MuLBSTA, and SmartCOP in the same picture. This graph will represent how your COV-Score discriminates against patients who are at high risk for mortality better than the other models.
A4.We added a ROC graph combining all three models (COV-Score, MuLBSTA, and SmartCOP) as supplementary figure (Figure S16 - ROC curve evaluation for prediction scores).

Reviewer 2 Report
The manuscript has been improved during the revision process. However, the authors should state in the limitations section that the model was created based on data from the first waves of the pandemic when other strains were prevalent, thus, applicability with the current strains could, theoretically, be an issue.
Author Response
We appreciate your time and work in reviewing our paper and providing valuable and insightful comments, leading to improvements in the current version. We hope the manuscript after the requested revisions meet your high standards.
Q1. The manuscript has been improved during the revision process. However, the authors should state in the limitations section that the model was created based on data from the first waves of the pandemic when other strains were prevalent, thus, applicability with the current strains could, theoretically, be an issue.
A1. Following mention has been added to Study limitation subsection: “The prediction model was created based on data from the first waves of the SARS-CoV-2 pandemic, and for applicability with current and future strains, further validation studies may be required.”
